# MANIFOLD CHARACTERISTICS THAT PREDICT DOWNSTREAM TASK PERFORMANCE

## ABSTRACT

Pretraining methods are typically compared by evaluating the accuracy of linear classifiers, transfer learning performance, or visually inspecting the representation manifold's (RM) lower-dimensional projections. We show that the differences between methods can be understood more clearly by investigating the RM directly, which allows for a more detailed comparison. To this end, we propose a framework and new metric to measure and compare different RMs. We also investigate and report on the RM characteristics for various pretraining methods. These characteristics are measured by applying sequentially larger local alterations to the input data, using white noise injections and Projected Gradient Descent (PGD) adversarial attacks, and then tracking each datapoint. We calculate the total distance moved for each datapoint and the relative change in distance between successive alterations. We show that self-supervised methods learn an RM where alterations lead to large but constant size changes, indicating a smoother RM than fully supervised methods. We then combine these measurements into one metric, the Representation Manifold Quality Metric (RMQM), where larger values indicate larger and less variable step sizes, and show that RMQM correlates positively with performance on downstream tasks.

## 1 INTRODUCTION

Understanding why deep neural networks generalise so well remains a topic of intense research, despite the practical successes that have been achieved with such networks. Less ambitiously than aiming for a complete understanding, we can search for characteristics that indicate good generalisation. Knowledge of such characteristics can then be incorporated into training methods and open more research avenues. These characteristics can also be used to evaluate and compare networks.

Arguably the most successful current theories of generalisation focus on the flatness of the loss surface at the minima (Hochreiter & Schmidhuber, 1997; Dziugaite & Roy, 2017; Dherin et al., 2021) (even though the most straightforward measures of flatness are known to be deficient Dinh et al. (2017)). Petzka et al. (2021) expands on this argument and shows that these methods correlate strongly with model performance, and reflect the assumption that the labels are locally constant in feature space. A thorough survey by Jiang et al. (2020) shows that some recent methods are, in fact, negatively correlated with generalisation.

To our knowledge, no theory looks at the structural characteristics of the learned Representation Manifold (RM) as a predictor for generalisation. We investigate whether structural characteristics in the RMs correlate with generalisation to task performance.

To illustrate the intuition behind our investigation, consider Figure 1, which represents two RMs, A and B. Assume that each RM is produced by the same architecture, trained on the same dataset; both have a flat minima but are trained with different methods. In the case of A, where the manifold is smooth, the sample representations of the Green class are, on average, closer to other Green class's points. Likewise, presentations of the Red class will, on average, be closer to other Red class's samples. On the other hand, if we consider RM B, there are chasms in the manifold that lead to some sample representations being closer to samples of the other class rather than samples of their own class, as illustrated in the blue patch.

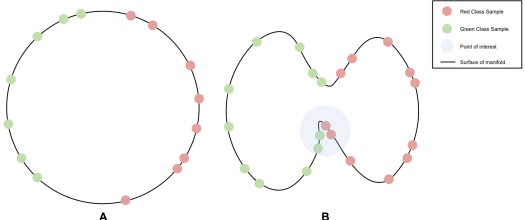

Figure 1: An illustration to give an intuitive understanding of why the structural characteristics of a RM should be considered a predictor of generalisation.

The purpose of this paper is to justify our claim that specific RM characteristics lead to generalisation. However, to do this, we must first define appropriate RM characteristics that reflect this intuition and show how to measure them.

**Contribution** This paper aims to show with enough empirical evidence that looking at the representation manifold (RM) structure is a good research direction for explaining generalisation in deep learning structures. Following these strong empirical results, future work will require a deeper theoretical investigation into our findings. Our contributions in this paper can then be summarised as defining a model-agnostic and straightforward framework to measure RM characteristics. Using this framework, we compare the RMs learned by encoders trained using supervised, self-supervised and a mixture of both methods on the MNIST and CIFAR-10 datasets. We then present a new metric that calculates the quality of a manifold for generalisation, the Representation Manifold Quality Metric (RMQM). We show that this metric correlates strongly with downstream task performance. These observations support our intuition on the characteristics of an RM that lead to generalisation.

## 2 RELATED WORK

**Representation learning** Some of the earliest work in representation learning focused on pre-training networks by generating artificial labels from images and then training the network to predict these labels (Doersch et al., 2015; Zhang et al.; Gidaris et al., 2018). Other techniques involve contrastive learning where representations from images are directly contrasted against one another such that the network learns to encode similar images to similar representations (Schroff et al., 2015; Oord et al., 2018; Chen et al., 2020; He et al., 2020; Le-Khac et al., 2020).

**Comparing representations from trained neural networks** Yamins et al. (2014); Cadena et al. (2019) compares how similar representations are by linearly regressing over the one representation to predict the other representation. The $R^2$ coefficient is then used as a metric to quantify similarity. This metric is not symmetric. Symmetrical methods compare representations from different neural networks by creating a similarity matrix between the hidden representations of all layers as was done in (Laakso & Cottrell, 2000; Kriegeskorte et al., 2008; Li et al., 2016; Wang et al., 2018; Kornblith et al.).

**Manifold Learning** The Manifold Hypothesis states that practical high dimensional datasets lie on a much lower dimensional manifold Carlsson et al. (2008); Fefferman et al. (2016); Goodfellow et al. (2016). Manifold learning techniques aim to learn this lower-dimensional manifold by performing non-linear dimensionality reduction. A typical application of these non-linear reductions is visualising high dimensional data in two-dimensional or three-dimensional settings. Popular techniques includeTenenbaum et al. (2000); Van der Maaten & Hinton (2008); McInnes et al. (2018). These techniques have been used in various studies to compare different learned representation manifolds (Chen et al., 2019; van der Merwe, 2020; Li et al., 2020; Liu et al., 2022).

**Comparing manifolds** To evaluate the performance of Generative Adversarial Networks, Barannikov et al. (2021) introduces the Cross-Barcode tool that measures the differences in topologies between two manifolds, which they approximate by the sampled data points from the underlying data distributions. They then derive the Manifold Topology Divergence based on the sum of the

lengths of segments in the Cross-Barcode. Zhou et al. (2021) also evaluates generative models by quantifying representation disentanglement. They do this by measuring the topological similarity of conditional submanifolds from the latent space. Shao et al. (2018) investigates the Riemannian geometry of latent manifolds, specifically the curvature of the manifolds. They conclude that having latent coordinates that approximate geodesics is a desirable property of latent manifolds. To our knowledge, there has not been a study done on measuring the manifold's structural characteristics based on small local alterations to the input data, applied to non-generative encoders.

**Predictors of generalisation**  Jiang et al. (2020) performed a large scale study of generalisation in deep learning, and we refer the reader to this work for a well-documented review. To our knowledge, there has been no work done on using the structure of the RM as a predictor of generalisation.

## 3  APPROACH

Describing all the details of a high-dimensional representation manifold (RM) is an impossible task; we can at best strive to find characteristics that summarise salient properties of the RM. When measuring these characteristics, one will therefore have a discrete view of the RM Barannikov et al. (2021), made out of the predicted representations from the input data. We propose measuring individual distance metrics for each representation of an input sample relative to representations of data close to it.

By staying in the neighbourhood of each representation, we can measure the surface surrounding that point using standard distance metrics, effectively walking on the local structure and measuring the size of each step relative to the change in input. By inspecting all these locally measured structures together, one describes the structure for the entire RM in terms of its local stability.

The caveat is that one requires representations in close proximity on the RM to do these measurements. However, practical RMs have many dimensions, implying that data points tend to be well separated, even if they originate from the same underlying class (Bárány & Füredi, 1988; Balestriero et al., 2021). We thus need to create these proximate representations artificially.

We do this by applying sequentially larger local alterations to the input data and computing the resulting representations. By increasing the size of the alteration, we step further on the RM surface and thus measure characteristics further away but still local for the magnitude of changes that we employ.

### 3.1  ALTERATION METHODS

Let an RM be represented by $\phi = f(\boldsymbol{X})$, where $f$ is a feature extractor and $\boldsymbol{X}$ is a dataset (e.g. input images). With $A$ representing a function that applies small local alterations to pixels in the image, each altered data point projected down to the RM is represented as $\phi_i = f(A_j(x_i))$, where $x_i \in \boldsymbol{X}$ and $A_j$ the $j$th iteration of the alteration function.

The two alteration methods we use in our experiments are the same methods Hoffman et al. (2019) used to evaluate the robustness of a Jacobian regulariser. We chose these methods because they result in either random local alterations or guided alteration, thus giving us different paths on the RM to evaluate.

**White noise injection.**  Here we alter each input image $x_i$ by adding an alteration vector randomly, $a$, with components independently drawn from a normal distribution with variance $\epsilon^2$, thus $a \sim \mathcal{N}(0, \epsilon^2)$. In order to increase the alteration strength, we increase $\epsilon$ from zero to one in 100 in equal steps, indexed by $j$. Thus, alteration $j$ for datapoint $x_i$ is given by $x_{ij} = [x_i + a_j]_{clip}$, where $a_j \sim \mathcal{N}(0, \epsilon_j^2)$ and $[.]_{clip}$ clips the image to be between zero and one.

**PGD attack.**  Whereas white noise injections will allow us to walk on the surface of an RM in random directions, altering the image in a way that deliberately aims to fool the trained function $f_\theta$ will allow us to walk in a direction influenced by decision boundaries on the RM. In this paper we will implement an extension of fast gradient sign method (FGSM) Goodfellow et al. (2015), namely projected gradient descent (PGD) (Madry et al., 2018). FGSM consists of adding a vector to

the original image, where this vector consists of the sign of the gradient for the loss functions with respect to the input image, scaled by a value $\epsilon_{FGSM}$. PGD iterates this process for several iterations. Calculating the $j$th alteration of $x_i$, represented by $x_{i,j}$ can be defined as

$$x_{i,j} = [x_{i,j-1} + \epsilon_{FSGM} \cdot (\nabla_{x_{i,j-1}} \mathcal{L}(\theta, x_{i,j-1}, y))]_{clip} \tag{1}$$

where $\mathcal{L}$ is the loss function for the relevant training method.

Given that the original target for these adversarial attacks was a network that classifies images. In order to then apply PGD attack to the triplet variants, we calculate the loss precisely as usual and then calculate the gradient with regards to the anchor image. When calculating the gradient for NT-XENT methods, we compare a non-augmented image with an augmented version and then calculate the gradient with respect to the unaugmented image. We apply the PGD attack for 30 iterations and save each iteration, with $\epsilon_{FGSM} = 2/255$.

### 3.2 MANIFOLD CHARACTERISTICS

Let $A_j$ be the $j$th iteration of an alteration method, where each successive iteration employs a stronger alteration. Also, let $\phi_{i,j}$ be the projected point on the RM produced by $f(A_j(x_i))$, where $x_i \in \boldsymbol{X}$.

**Average distance moved.** The first characteristic we measure is the total of the normalised Euclidean distances between the original point, $\phi_{i,0}$, and each altered point, $\phi_{i,j}$, where $\phi$ is the normalised vector. The average distance moved for image $x_i$ is represented as $\frac{1}{J} \sum_j \|\phi_{i,0} - \phi_{i,j}\|_2$, where we can then average over each point to find the average distance moved for a given RM and alteration. Finally, the average over images is

$$D(\phi, A) = \frac{1}{NJ} \sum_i^N \sum_j^J \|\phi_{i,0} - \phi_{i,j}\|_2 \tag{2}$$

where $N$ is the number of images considered and $J$ is the number of alterations. $D(\phi, A)$ thus indicates how robust the RM is to alterations $A$.

**Average distance spikes.** We also measure the relative change in distance between successive alterations. These relative distance changes are measured both with respect to the original representation and relative to each previous alteration. We average the magnitudes of these changes as we are not interested in the direction of the change to gauge how smooth an RM is. To understand how relative changes relate to smoothness, recall that we only apply alterations that keep us close to the given data points. Therefore, we can only have big spikes if the RM contains significant chasms or bumps (small alterations in input data should result in constant distance increases if the RM is smooth).

In order to calculate these relative changes for a single representation, refer to Equation (3) which calculates the relative change according to the original representation, $D_{RC}$ and Equation (4) which calculates the relative change according to the distance between the previous alterations $P_{RC}$. In both equations, $d()$ is a distance function. In order to get the overall metrics, we average the values over all data points.

$$D(\phi_i, A)_{RC} = \frac{1}{J} \sum_{j=1}^J \left| \frac{d(\phi_{i,0}, \phi_{i,j}) - d(\phi_{i,0}, \phi_{i,j-1})}{d(\phi_{i,0}, \phi_{i,j})} \right| \tag{3}$$

$$P(\phi_i, A)_{RC} = \frac{1}{J} \sum_{j=2}^J \left| \frac{d(\phi_{i,j-1}, \phi_{i,j}) - d(\phi_{i,j-1}, \phi_{i,j-2})}{d(\phi_{i,j-1}, \phi_{i,j})} \right| \tag{4}$$

## 4 COMPARING DIFFERENT TRAINING METHODS

We now measure the characteristics defined in Section 3 for five different training methods applied to two different encoders and datasets. These five methods range from purely unsupervised to fully

supervised. By comparing these five methods, we discover common properties for related methods, thereby gaining a better understanding of the learned RMs. For further details on these training methods, please refer to Appendix C.

To ensure that each learned RM can be compared fairly to each other, all image alterations are exactly the same for each method when calculating the manifold characteristics. We report in the rest of this section using the average results over both the Adam and SGD trained encoders, as the results were similar for both.

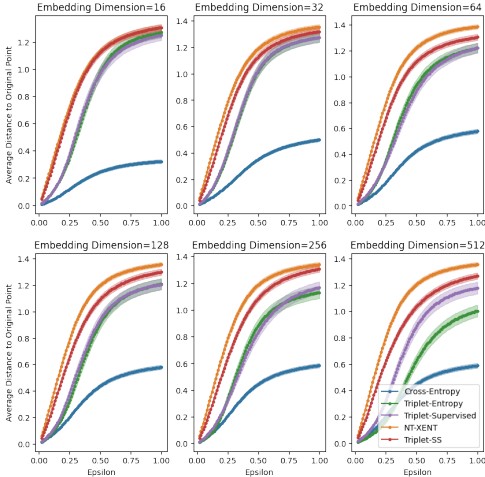

Figure 2: The normalised Euclidean distance between the original MNIST digit and the same digit altered by our white noise injection method. We perform this measurement for each embedding dimension and digit in the test set. We then calculate the standard error shown with the mean results.

## 4.1 MEASURING DISTANCES

Figure 2 shows the average normalised Euclidean distance to the original MNIST digits as we increase the amount of alteration applied to each digit. Here, $A$ is the white noise injection alteration. As the embedding dimension increases, the self-supervised methods move further from the original point than the supervised signal methods. We also notice that the Cross-Entropy encoder's average distance away from the original representation stays very low, with the Triplet-Entropy and Triplet-Supervised encoders falling in between. We suspect this is because they contain both supervised and unsupervised signals in the training process.

When $A$ is the PGD attack, we see the same pattern emerging for the CIFAR-10 encoders, shown in Figure 3. Here though, we can see a much more significant difference between self-supervised methods and methods containing a supervised signal: the NT-XENT and Triplet-SS measurements grow to have much larger values.

In Table 1 we summarise the $D$ values, which is calculated using Equation (2), for the MNIST encoders. We average over embedding dimensions and calculate the standard deviation for each method. The common trend among both alteration methods is that NT-XENT and Triplet-SS alterations always move farther away from the original representation than the other methods. We can also see that the total distance moved decreases from self-supervised to pure supervised learning methods.

These results indicate that the encoder is more robust to minor perturbations (as measured by the distance moved from the original image) if the training method contains a strong supervised signal.

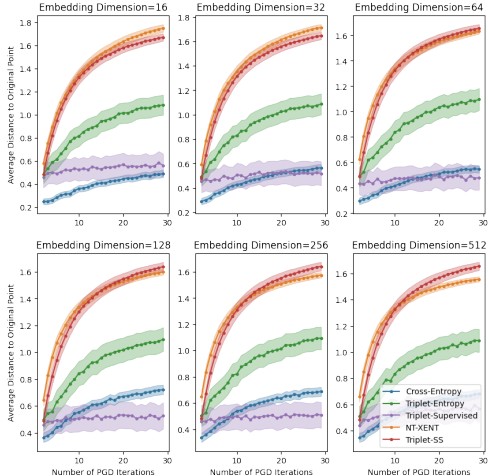

Figure 3: The normalised Euclidean distance between the original CIFAR-10 image and the same image altered by sequential PGD attack iterations.

Table 1: The average distance each points moves relative to the original point for both the white noise injection and PGD Attack alterations for the MNIST encoders. The results are averaged over both the embedding dimension of an encoder and the optimiser used.

| METHOD | NOISE | PGD |
|---|---|---|
| CROSS-ENTROPY | 0.33±0.15 | 0.21±0.08 |
| TRIPLET-ENTROPY | 0.73±0.11 | 0.33±0.08 |
| TRIPLET-SUPERVISED | 0.77±0.06 | 0.38±0.10 |
| TRIPLET-SS | 0.93±0.13 | 0.66±0.17 |
| NT-XENT | 1.01±0.04 | 0.68±0.10 |

## 4.2 MEASURING SPIKES

Following the same steps as in Section 4.1, we now study the relative change in distances measured. The relative change in distance to the original representation, plotted against the amount of PGD attack iterations, is shown in Figure 4.

We see a reversal of the graphs in Section 4.1: the self-supervised methods start with high relative changes, which decrease rapidly. Methods containing a supervised signal have larger spikes and error bands. All this indicates a less smooth journey in the RM between alterations for the supervised methods.

The same trend is present for CIFAR-10 models when we inject white noise. Here though, the Triplet-Supervised method is unstable for several values of the embedding dimension, whereas the NT-XENT and Triplet-SS again have the smallest spikes.

Table 2 shows the overall average values for each spike metric for each method. NT-XENT and Triplet-SS have the smallest spikes for both forms of alteration, whereas Triplet-Supervised results in very non-smooth RMs.

Self-supervised methods therefore learn structures in which a step in most directions, at most locations, induces steps of similar size on the RM. That is, these self-supervised methods have smoother RMs than the other methods.

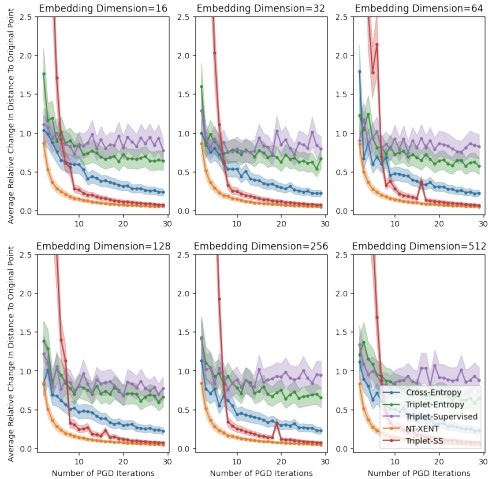

Figure 4: The relative change in distance to the original point plotted against the number of PGD iterations for the encoders trained on the MNIST dataset.

Table 2: The average change in the distance each point moves relative to the original point, compared against the previous alteration's distance. Results are calculated for both the white noise injection and PGD Attack alterations for the CIFAR-10 encoders. The results are averaged over both the embedding dimension of an encoder and the optimiser used.

| METHOD | NOISE | PGD |
|---|---|---|
| CROSS-ENTROPY | 0.11±0.03 | 0.34±0.04 |
| TRIPLET-ENTROPY | 0.18±0.04 | 0.89±0.35 |
| TRIPLET-SUPERVISED | 0.86±0.85 | 2.97±1.61 |
| TRIPLET-SS | 0.06±0.01 | 0.06±0.04 |
| NT-XENT | 0.04±0.01 | 0.10±0.01 |

## 5    REPRESENTATION MANIFOLD QUALITY METRIC

In Section 4 we showed empirically that an RM learned by self-supervised methods has a structure that has the following property: When moving in any direction on the surface of RM, it will result in a relatively large displacement, but these displacements are on average the same size no matter where or in what direction a step is taken. The opposite is true with methods containing a supervised signal: moving in the surface results in smaller displacements, but those displacements are significantly more variable in size.

In order to determine which of these two groups of characteristics are more desirable for downstream tasks, we combine these characteristics into one metric, the Representation Manifold Quality Metric (RMQM).

With this single metric describing an RM, we can perform various downstream tasks with our encoders and see how the performance correlates with the value of the RMQM. We define the RMQM as

$$RMQM = \ln\left(1 + D + D_{PC}^{-1} + P_{PC}^{-1}\right) \tag{5}$$

Here $D$ is the average distance moved measured relative to the original representation, $D_{PC}$ is the relative change in distance between each subsequent alteration and the original representation and $P_{PC}$ is the relative change of the distances between altered representations, as defined in Equations (2) to (4).

We apply the natural logarithm to scale the values, and we add one to ensure we do not have any negative values.

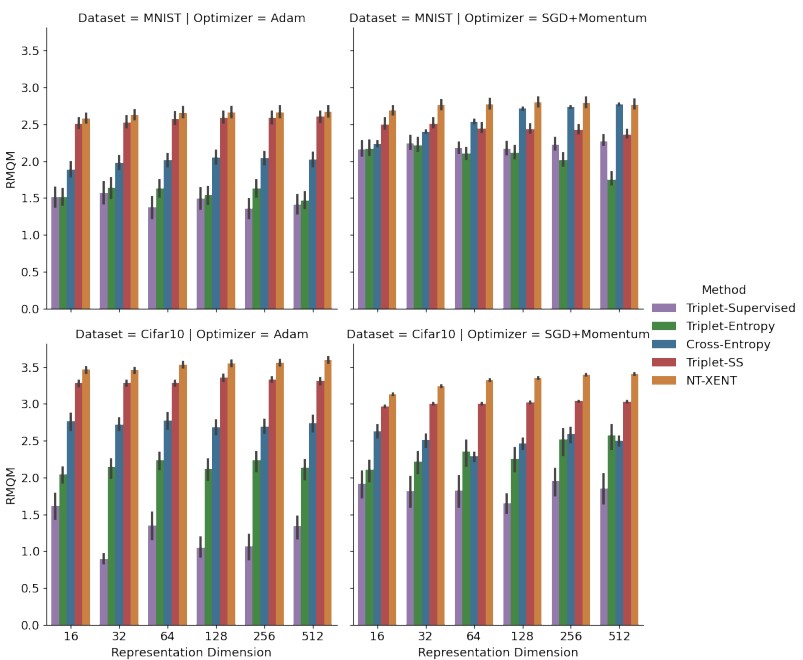

Figure 5: RMQM for white noise injections, over every embedding dimension for each encoder.

Thus, RMQM is designed to yield large values for relatively smooth RMs with relatively large sensitivity to changes in the input. Below, we only interpret the RMQM score when $A$ is the white noise injection alteration, since the method dependence of the PGD alterations complicates our ability to compare the various methods.

In Figure 5 we show the RMQM score for each of our encoders, with $A$ being white noise injections. For the MNIST encoders trained using SGD and Nesterov momentum, as the embedding size increases, the RMQM for Cross-Entropy overtakes Triplet-SS, indicating that the RM is more similar in this setup to one produced by NT-XENT trained encoders. In the other cases, there is an overall trend for the NT-XENT and Triplet-SS encoders to have the highest RMQM, followed by Cross-Entropy and then lastly, Triplet-Entropy and Triplet-Supervised.

## 5.1 CORRELATION BETWEEN RMQM AND DOWNSTREAM TASKS.

In order to find what RM characteristics are desirable, we measure how RMQM correlates with downstream task performance. If we find a strong positive correlation, an RM with a smooth structure and large displacements is desirable. If there is a strong negative correlation, then an RM that contains chasms and bumps and small displacements is desirable.

We define the task performance as the normalised test accuracy of a K-Nearest Neighbour (KNN) model, with only one nearest neighbour ($K = 1$), trained on the representations created by the encoder, as well as the normalised test accuracy when doing a linear probe on the final layers of the network. We believe these are appropriate measures of task performance as most use cases today utilise representations. We also use these tasks, as these task do not change the learned RM structure, as would be the case when fine-tuning the encoders on a new dataset with Cross-Entropy.

The MNIST encoders will be tested on the OMNIGLOT (Lake et al., 2015) and the KMNIST (Clanuwat et al., 2018) datasets, with the CIFAR-10 encoders being tested on the Caltech-101 (Fei-Fei et al., 2004) and CIFAR-100 (Krizhevsky et al., 2009) datasets. We believe these datasets provide a large enough semantic gap towards the original datasets, as we do not measure for extreme generalisation to any dataset. We exclude the Omniglot dataset results for the linear probe performs tasks as they perform poorly across all model variations (we believe the task is too hard for probing), which skews the results heavily.

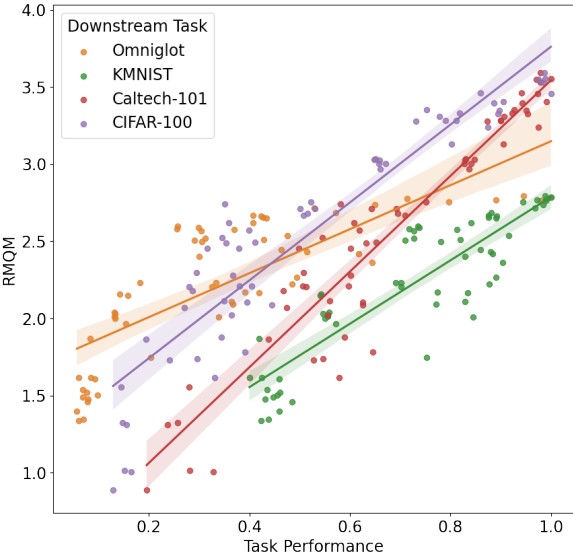

Figure 6: RMQM versus the normalised KNN task performance. The dots represent each encoder's performance and RMQM score, colour coded by the downstream dataset. As one can see, there is a strong positive correlation between RMQM and downstream task performance

Figure 6 visualises RMQM versus the downstream KNN task performance, where each dot refers to a differently trained model. We find that RMQM correlates positively to the KNN task with a coefficient of $0.75$. These results also stay true for larger values of $K$. We also find a positive correlation to the linear probing results with a coefficient of $0.78$. Combining both of the tasks, there is a positive correlation of $0.72$. This justifies our statement that RM characteristics are essential. In other words, when vector search is the downstream task performance, an encoder that learned an RM with a smooth structure and large displacements will tend to perform well on downstream search tasks. To gain an intuition for why this can be, please refer to Appendix B, where we give our intuitive explanation.

## 6 CONCLUSION

We propose a framework to measure the characteristics of learned representation manifolds (RM). We measure the characteristics by applying sequentially stronger local alterations to the input data and measuring how these altered representations move relative to the original representation and the successive alterations. We show that self-supervised learning methods learn RMs in which motion in any direction on the surface will result in relatively large displacements. However, these displacements are relatively similar no matter where or in what direction a step is taken.

To identify RM characteristics related to good downstream task performance, we combine our measurements into a single metric, the Representation Manifold Quality Metric (RMQM). RMQM is designed to yield large values for relatively smooth RMs with relatively large sensitivity to changes in the input. We then measure the downstream task performance for several tasks and find a strong positive correlation with RMQM. This strong correlation indicates that the structure of a learned manifold is another strong predictor for the generalisation of neural networks. This also shows that self-supervised methods lead to state-of-the-art performance due to the underlying RM structure, which is sensitive to alterations in the input, utilising a relatively smooth manifold.

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

## A    RMQM ABLATION STUDY

To find which features of the RMQM formula contribute the most to predicting the overall downstream task performance, we performed an ablation study where we compared different combinations of inputs to the RMQM formula described in eq. (5).

When the eq. (5) consists only of the average distance moved, $D$, there is a very low correlation of only $0.27$ to downstream task performance. We then looked at when only spikes are considered, $D_{RC}$ and $P_{RC}$, and found that the correlation was now $0.72$, just below the reported $0.75$ when everything is combined. We also measured all the remaining combinations and found that the correlation ranged between $0.71$ and $0.74$. This indicates that even though the spikes can be used to measure performance, adding the average distance increases the performance of the RMQM score.

Finally, we optimised the weights associated with each variable we found that the following combination, shown in Equation (6), maximises the correlation at $0.784$.

$$RMQM = \ln\left(1 + 0.95D + 0.74D_{PC}^{-1} + 0.11P_{PC}^{-1}\right) \tag{6}$$

These results show that one can not only look at distance moved on its own and that having small spikes contribute significantly to generalisation. Nevertheless, spikes must be considered along with distance for the best predictive capability.

These results indicate that previous work on model robustness still holds, as $D$ contributes less than spikes. However, robustness and RMQM complement each other in predicting the generalisation of neural networks.

## B    INTUITIVE EXPLANATION OF RMQM CONCLUSIONS

To help better understand these perhaps unintuitive results, consider the case when we calculate the RMQM using white noise alterations. Take the MNIST encoders as an example, and consider a specific MNIST image (any digit). When applied to this digit, a random vector exists that will transform it into one of the Omniglot characters, given a large enough alteration. In general, the variants of this Omniglot character will differ from that digit image by similar noise vectors. Thus when we encode this Omniglot character image and its variants using a self-supervised trained encoder, the KNN models can accurately identify new images because, from the perspective of the RM, these new characters correspond to the noise-altered version of our original MNIST digit. On the RM, this new character and its variants are projected with a similar step size away from the original MNIST digit. Due to the similar noise vector added, these projections are also in a similar direction. These projected characters are also close because there are few chasms or bumps a projection can land on, allowing a nearest neighbour search to perform well.

## C    DETAILS ON TRAINING METHODS

The first method we investigate employs encoders trained with vanilla supervised learning with Cross-Entropy loss, where we then take the second to last layer output as the representations. We also use the SimCLR method introduced in Chen et al. (2020). This method compares two augmented versions of the same image and brings their representations closer together using the Normalised Temperature-scaled Cross-Entropy (NT-XENT) Loss (Sohn, 2016). Along with this, we trained encoders using two different implementations of Triplet-Loss (Weinberger et al., 2006; Schroff et al., 2015). We believe these techniques represent the major families of training techniques and provide enough information on how different techniques learn different RM structures. In a future paper, we propose that a full-scale investigation be performed on most training techniques found in current literature.

We mine the triplets in a supervised manner for the first implementation using the image labels. We apply the SimCLR method for the second implementation, replacing NT-XENT with Triplet loss. We refer to the former method as Triplet-Supervised going forward and the latter as Triplet-SS. We do this to see the effect of the indirect supervised signal on the method. Lastly, to see the effect of

directly combing a contrastive signal with a supervised signal, we combine Triplet-Loss with Cross-Entropy loss, as was also done in van der Merwe (2020). Here the second to last layer's outputs are fed into the Triplet-Loss function, where the triplets are mined using the same labels used for the Cross-Entropy loss, which takes as input the logits produced by the last layer.

We apply these methods to an altered version of the LeNet-5 architecture introduced in LeCun et al. (1998), trained on the MNIST dataset (LeCun et al., 1998) and a Resnet-18 (He et al., 2016) trained on the CIFAR-10 dataset (Krizhevsky et al., 2009). We train our encoders with six different embedding sizes, ranging from 16 to 512 in powers of two. We also train with two different optimisers, namely Stochastic Gradient Descent (SGD) with Nesterov Momentum (Sutskever et al., 2013) and Adam (Kingma & Ba, 2015).

For SGD with Nesterov momentum, we set the learning rate to $0.001$ and momentum to $0.9$. Our learning rate is also $0.001$ with the default PyTorch hyperparameters for Adam. We train the CIFAR-10 encoders for 100 epochs and the MNIST encoders for seven epochs.

