# OpenReview forum: "Manifold Characteristics That Predict Downstream Task Performance"
_ICLR.cc/2023/Conference — Submitted to ICLR 2023_

### Official Review · Reviewer_xaAq · 2022-10-24

**Confidence:** 3
**Correctness:** 3
**Technical Novelty And Significance:** 2
**Empirical Novelty And Significance:** Not applicable
**Recommendation:** 5

**Clarity, Quality, Novelty And Reproducibility:**

The text is quite clear (but the overall presentation of the paper is poor) and I think it will be easy to reproduce the work done in this paper.

**Strength And Weaknesses:**

Strength:
- Novelty. This paper relates prediction performance of downstream tasks with the characteristic of learnt manifold, which I think is quite interesting.

Weakness:
- Presentation of the paper. Too much white spaces and small text in the figures. Captions do not sufficiently describe the figures.
- "trained with different methods" in the 4th paragraph in the intro is a little confusing. I thought "different methods" would refer to different optimization scheme but later it turns out that they refer to different tasks.
- Perhaps a figure of the overall pipeline of the method will help a reader understand the ideas and aims of this paper.
- There is a big jump going from Fig 2 to Fig 3 as two nubs are turned at the same time, i.e., dataset and perturbation method. It is difficult to tell which factor is causing
- The authors propose a metric RMQM to measure the quality (e.g., smoothness) of the learned manifold. There must be other metrics, I am sorry but I am not sure which ones though (but at least the authors have done literature survey on manifold learning and comparison methods), but there is no way to tell that RMQM is the right one.
- Lack of theory. While the paper is proposing interesting theories, they are only empirically validated with very limited settings, and it is not certain if the same observation will extend to more complicated experiments especially that may contain exhaustive local minima.

**Summary Of The Paper:**

This paper investigates properties of trained manifolds with different learning methods on simple image data. For this, the authors propose data augmentation method and define metrics that lead to quantification of the quality of the overall manifold. They show that the proposed metric, i.e., RMQM, has high correlation with performances of several downstream prediction tasks.

**Summary Of The Review:**

The authors provide an interesting aspect of training machine learning models for interpretability. However, the ideas are not sufficiently validated or proved, and they should be better presented.

---

### Official Review · Reviewer_s4yj · 2022-10-24

**Confidence:** 5
**Correctness:** 1
**Technical Novelty And Significance:** 2
**Empirical Novelty And Significance:** 2
**Recommendation:** 3

**Clarity, Quality, Novelty And Reproducibility:**

The term metric is not used appropriately. Indeed, distance, metric and displacement are different entities.

A distance metric is defined precisely in mathematics for different spaces. According to the well-known definitions of metrics employed for Euclidean spaces and nonlinear manifolds, (3) and (4) are not metric, therefore, RMQM is not a metric.

To resolve this issue, please either define your distances and metrics more precisely clarifying the differences between them, prove metric properties of the proposed functions, or use some other terms for the proposed entities.

Another major problem is comparison of points on manifolds. According to the discussions given in the motivation sections (e.g. Fig. 1), I suppose that nonlinear manifolds instead of linear Euclidean spaces are considered as RMs in the paper. Then, it is not clear why points on nonlinear manifolds are compared using Euclidean distances.

In addition, experimental analyses should be improved according to the proposed claims. First, additional datasets, methods and models should be explored for different vision tasks in addition to the classification tasks, and also for other AI tasks such as NLP and speech recognition. Second, the paper claims some results for self-supervised learning methods. Then, this claim should be verified using state-of-the-art self-supervised learning methods as well. If the experimental analyses cannot be extended, then some of the claims should be mathematically explored.

Some typo:
includeTenenbaum → include Tenenbaum
Zhou et al. (2021) also evaluates → Zhou et al. (2021) also evaluate

**Strength And Weaknesses:**

The paper addresses an important problem of deep learning.

However, there are various major and minor problems with the paper while addressing this problem. Briefly, there are unclear definitions and proposed methods are not employed correctly following common definitions and structures of manifolds. In addition, experimental analyses should be improved.

**Summary Of The Paper:**

This paper proposes a framework and new methods to measure and compare different representation manifolds (RMs) to explore various pretraining methods. The analyses show that some self-supervised methods learn an RM where alterations lead to large but constant size changes, indicating a smoother RM than fully supervised methods. These measurements are integrated to develop a new  method called the Representation Manifold Quality Metric (RMQM), which is used to explore relationship between accuracy of models and RMs on downstream tasks.

**Summary Of The Review:**

The paper addresses an important problem of deep learning. However, there are various major and minor issues with the paper. Therefore, the paper is not ready for publication without fixing these issues.

---

### Official Review · Reviewer_jVym · 2022-10-25

**Confidence:** 4
**Correctness:** 2
**Technical Novelty And Significance:** 3
**Empirical Novelty And Significance:** 3
**Recommendation:** 6

**Clarity, Quality, Novelty And Reproducibility:**

In terms of novelty it seems strange to me that "there has been no work done on using the structure of the RM as a predictor of generalisation". Although, studying the perturbations may be novel. This work for one uses the compactness of the representation to predict generalization https://arxiv.org/abs/2012.02775 and I suggest the authors read the proceedings of this workshop to identify relevant methods to contrast theirs with https://sites.google.com/view/pgdl2020/resources Some of these methods can be used as baselines to help understand the significance of the perturbation approach.

Also, when the models were the models selected using early stopping? One experiment that could be interesting (not saying you need it for publication) is if you compute the RMQM at each epoch. One would expect to see a similar trend to the validation loss. An application of this would share the motivation of this work: https://arxiv.org/abs/1703.09580 which aims to not use a validation dataset.


**Strength And Weaknesses:**

Strengths: The paper is well organized. The argument to predict generalization is clear. The proposed metric is explained clearly.

Weaknesses: I would have preferred to see the evaluation performed on larger datasets and not just small toy datasets. The support for the claim in the paper is solely empirical so this should be expanded on more. I also believe there should be more comparisons to existing work.


**Summary Of The Paper:**

The paper presents a score for evaluating the representational space of a model to estimate the potential generalization performance. An experiment with this method across a few datasets provides evidence to support this claim.

**Summary Of The Review:**

I really enjoyed reading this paper and I think the authors are doing some great work and have great ideas. However, I would prefer that the claims are supported with more complex datasets as well as contrasted with existing baselines. I believe the authors will find some from the publications in the PGDL2020 workshop that will demonstrate the potential improvement of their method.

---

### Decision · Program_Chairs · 2023-01-20

**Decision:**

Reject

**Justification For Why Not Higher Score:**

Two out of three reviewers recommend rejection and I do not have enough grounds to go against their recommendation.

**Justification For Why Not Lower Score:**

N/A

**Metareview: Summary, Strengths And Weaknesses:**

Summary: This paper presents a metric for evaluating a model's representation manifold's ability to generalize to downstream tasks.

Strengths: The paper is well written and addresses an interesting problem.

Weaknesses: The experimental validation would benefit from larger datasets and more extensive comparison to other work.